# The effects of group adaptation on task performance: An agent-based approach

**Darío Blanco-Fernández**[1]*, **Stephan Leitner**[2], **Alexandra Rausch**[2]

**1** Digital Age Research Center, University of Klagenfurt, Klagenfurt, Austria, **2** Department of Management Control and Strategic Management, University of Klagenfurt, Klagenfurt, Austria

* dario.blanco@aau.at

**Data Availability Statement:** All source code and dataset files are available at Figshare (https://figshare.com/projects/The_effects_of_group_adaptation_on_task_performance_An_agent-based_approach/174876) and Gitlab

## Abstract

Organisations rely upon group formation to solve complex tasks, and groups often adapt to the demands of the task they face by changing their composition periodically. Previous research has often employed experimental, survey-based, and fieldwork methods to study the effects of group adaptation on task performance. This paper, by contrast, employs an agent-based approach to study these effects. There are three reasons why we do so. First, agent-based modelling and simulation allows to take into account further factors that might moderate the relationship between group adaptation and task performance, such as individual learning and task complexity. Second, such an approach allows to study large variations in the variables of interest, which contributes to the generalisation of our results. Finally, by employing an agent-based approach, we are able to study the longitudinal effects of group adaptation on task performance. Longitudinal analyses are often missing in prior related research. Our results indicate that reorganising well-performing groups might be beneficial, but only if individual learning is restricted. However, there are also cases in which group adaptation might unfold adverse effects. We provide extensive analyses that shed additional light on and help explain the ambiguous results of previous research.

## 1 Introduction

Change is often unavoidable and even necessary. As [1, 2] note, groups often change their composition to *adapt* to the task conditions over time. By adapting, groups aim at improving task performance [3, 4]. Negative consequences of adaptation, however, might also exist [5]. This paper takes up these ideas and examines the effects of *group adaptation* to the task environment on performance.

There are four main strands of relevant research on group adaptation. First, conceptual research is concerned with the temporal aspects of groups, including the emergence of efficient group compositions [1, 2, 6]. Second, there is research on the relationship between group adaptation and learning [7–11]. Third, some research focuses on the effects of group adaptation on creativity and innovation [12–15]. Finally, the fourth strand of research studies how groups respond and adapt to internal and external changes [3–5]. Prior research, however, has not extensively studied how group adaptation—understood as a process of changing a group's composition—affects task performance.

(https://gitlab.aau.at/dablancofern/nk-model-for-dynamic-groups).

**Funding:** The author(s) received no specific funding for this work.

**Competing interests:** The authors have declared that no competing interests exist.

We place our research in this gap and consider task complexity and individual learning as moderating factors in the relationship between group adaptation and performance. Previous research provides considerable evidence that task complexity—in terms of the number and pattern of the interdependencies between subtasks—strongly affects performance [16–19]. [20] provides some examples to illustrate task complexity. A holding company might be formed by several departments which invest independently. The overall investment pro-gramme of the firm is a task of low complexity, as there are few interdependencies between the investment units. By contrast, designing and building a car engine is a moderately complex task. Concerning learning, [21] argues that there might be an interaction between individual learning (i.e., creating knowledge within a group) and changing a group's composition (i.e., absorbing knowledge outside of the group by attracting new members), since, from the group's perspective, both approaches result in new knowledge. Consequently, we expect moderating effects of task complexity and learning in the relationship between group adaptation and performance.

In this paper, we propose an agent-based approach to study this relationship and the mod-erating effects. Our approach differs from most of the prior research on group adaptation, which usually employs experimental, survey-based, or fieldwork methods [5, 22]. In contrast to these methods, agent based-modelling does not involve the direct observation of real-life decision-makers. It is a computational technique which allows researchers to model multiple heterogeneous agents who interact with and within an environment [23].

There are several reasons why this method is suitable for studying the effects of group adap-tation on performance. Agent-based modelling and simulation allows the researcher to control the research conditions and, consequently, to study causal relationships between the variables of interest [24]. Additionally, since agent-based approaches employ computer simulations, it is possible to investigate large sets of variables and to control for simultaneous variations in the variables and the agents' behaviour [23, 24]. In this regard, experimental, survey-based, and fieldwork methods have several disadvantages. Experimental research allows the researcher to closely monitor the research conditions, too, but this tight control is somewhat problematic. In their research on group adaptation, [12, 13, 25, 26] note that experimental methods limit the number of variables that can be simultaneously studied. Also, the experimental design lim-its the variation in the variables [24]. Regarding survey-based and fieldwork methods, the researcher generally has little to no control over the research conditions [24]. [10, 27], who sur-vey managers to study group adaptation, acknowledge that this might result in problems such as data inconsistency, personal biases, and measuring errors. Similar problems are reported by researchers who employ fieldwork methods, such as [8, 9, 28]. [28] identify unpredictability in the research environment as the main reason behind these problems.

Besides the number of variables and their variations, prior research is often limited in the temporal analysis of the variables. [5] point out that longitudinal studies of group adaptation are rare in the literature. In general, survey-based and experimental methods do hardly allow for longitudinal analyses [24]. [10, 11] bring forward the argument that longitudinal studies could reduce the problems of surveys and allow researchers to better understand the variables of interest, but that this is particularly challenging and costly [24, 29]. Additionally, the tempo-ral limitations of experimental research entail that researchers can study group adaptation only by measuring the effects of one-off changes in group composition [12, 25, 26]. [26] sug-gest that further valuable insights can be gained by studying group adaptation in the long term. In contrast to, in particular, empirical research methods, agent-based modelling and simulation allows for a longitudinal study of multiple variables [24]. Since the method involves computer simulations, it is easier for the researcher to establish a long-term perspective on the question of interest. Consequently, prior research which employs agent-based modelling and

simulation, such as [17–19, 30–32], often takes temporal aspects into account and differentiates between short- and long-term analyses. By employing such an agent-based approach for our research objectives, we aim to fill the research gap that [5] highlight regarding the lack of prior longitudinal research on group adaptation.

Finally, experimental and fieldwork methods are often criticised for not allowing to draw general conclusions from the results [24], e.g., because of the tight conditions under which experiments are conducted [25, 26]. Fieldwork methods are not subject to such experimental constraints, but their results are strongly context-based and might therefore compromise the results' generalisation, too [8, 9, 28]. By contrast, the results derived from survey-based and agent-based approaches are easier to generalise [24]. Both methods avoid the tight constraints of experimental research and the contextual focus of fieldwork. However, they differ in how to draw general conclusions from the results. A particular strength of survey-based methods is that they allow to study real-life decision-makers in different fields and contexts. This facilitates generalisation [10]. Although the lack of direct observation of real-life decision-makers is a major disadvantage of agent-based modelling and simulation, agent-based modelling and simulation allows for generalisation, too, because a large set of variables and values is explored [24]. It is even easier to draw general conclusions, if the researcher employs a structured agent-based model which has been contrasted with other research techniques, such as the *NK* framework [24, 33, 34].

Against the background of the methodical advantages and disadvantages, we aim to contribute to the literature by employing an agent-based approach to study the effects of group adaptation on performance. For this purpose, we define our research questions as follows:

(i). What is the effect of group adaptation on task performance?

(ii). How does individual learning moderate the relationship between group adaptation and task performance?

(iii). How does complexity moderate the relationship between group adaptation and task performance?

To the best of our knowledge, the proposed agent-based approach is a novel application of simulation-based methods to study group adaptation. The approach might help overcome some of the limitations of previous research in this field and shed light on some of the ambiguous results regarding the effects of group adaptation of performance revealed in previous research.

The remainder of this paper is organised as follows: Section 2 provides the background for group adaptation, and Section 3 introduces the agent-based model. The results and their discussion are presented in Section 5. Section 6 concludes the paper.

## 2 Background

According to [1, 2, 6], organisations increasingly rely on *team*-based structures for their operations. In this context, the temporariness of teams as groups is a crucial aspect. [6] claim that groups usually change their composition after completing a task. Following this argument, [1, 2] state that these changes repeatedly occur in response to the task's demands. Consequently, one aim of group adaptation is to improve task performance by keeping the group composition up with the current task requirements [3, 4].

Prior research has found positive effects of group adaptation on task performance [4, 5]. However, researchers also suggest that more focus should lie on the so-called *dark side* of group adaptation, i.e., on possible adverse effects of changes in a group's composition on task

performance [5]. According to prior research, changes in group composition might result in *(i)* increasing the creativity and the diversity of solutions employed to solve a task [12–14] and *(ii)* integrating the best-available experts within the group's ranks [10], but also in *(iii)* offsetting the positive effects that individual learning has on task performance [7, 11]. Consequently, this line of research argues that any decision involving group adaptation results in a trade-off, and groups should balance its advantages, i.e., *(i)* and *(ii)*, and disadvantages, i.e., *(iii)* [5, 10].

However, how group adaptation translates into task performance is not extensively addressed in the literature. [12] employ an experimental framework to study whether replacing one group member is associated or not with improvements in task performance. In another study, [27] employ survey-based methods to examine the interrelations between groups with a stable composition, group learning, and task performance. Additionally, [35] empirically study how changes in the composition of groups dedicated to video game development affect the critical reception and the sales of the published games. While differing in their methods, these three studies arrive at a similar conclusion: Changes in group composition do not improve the performance of already high-performing groups [12, 27, 35].

Prior research sometimes considers it beneficial to fix the composition of groups formed to solve complex tasks [2]. In contrast, other lines of research regard adaptivity as beneficial and claim that a greater focus should be put on better understanding the temporal aspects of groups, and in particular of the adaptation of groups [1, 2, 5, 6, 36]. We place our research in this tension and aim at shedding additional light on these ambiguous recommendations.

## 3 The model

Following the research questions outlined in Section 1, we propose an agent-based model in which agents form a group to collectively solve a complex task. The objective of this model is to asses how certain elements, such as group adaptation, individual learning, and complexity influence task performance—the observable variable of the model—over an extended period of time. The model is formed by four building blocks, which correspond to the sequence of events of the model, see Fig 1. The following subsections outline these four building blocks, reflecting this sequential order. The initialisation phase occurs right before the first time step $t = 1$. In this initial phase, the parameters and the agents of the model are set up (see Section 3.1). After the initial set-up has been completed, the signal-based group formation process takes place (see Section 3.2). Depending on the parameter configuration, this process may be repeated periodically, allowing us to assess how adapting the group composition more or less frequently affects task performance. Once the group is formed, its members choose a particular solution to the task based on their own knowledge and estimations (see Section 3.3). The solution chosen at time $t$ is translated into a particular group performance at that time step, and team members experience the resulting utility. Finally, at the end of each time step, agents may learn about the task by sequentially exploring the solution space (see Section 3.4).

Table 1 provides an overview of the key variables of the model and their values.

The model has been implemented in Python 3.7.4. We ran the code on the Spyder software version 5.0.0 using two different devices: a laptop with 16GB of RAM and a 1.90 GHz Intel Core i7 Processor and a laptop with 8GB of RAM and a 3.30 GHz AMD Ryzen 5 5600H with Radeon Graphics processor. Every scenario consists of $\Phi = 1,500$ simulation rounds of $T = 100$ periods each, where each period is denoted by $t \in \{1, \ldots, T\}$ and each simulation round by $\varphi \in \{1, \ldots, \Phi\}$. Simulating each scenario takes between 15 and 90 minutes, depending on the device employed and the parameter settings. The code as well as a structured model description following the ODD+D protocol introduced in [37] are provided here and here.

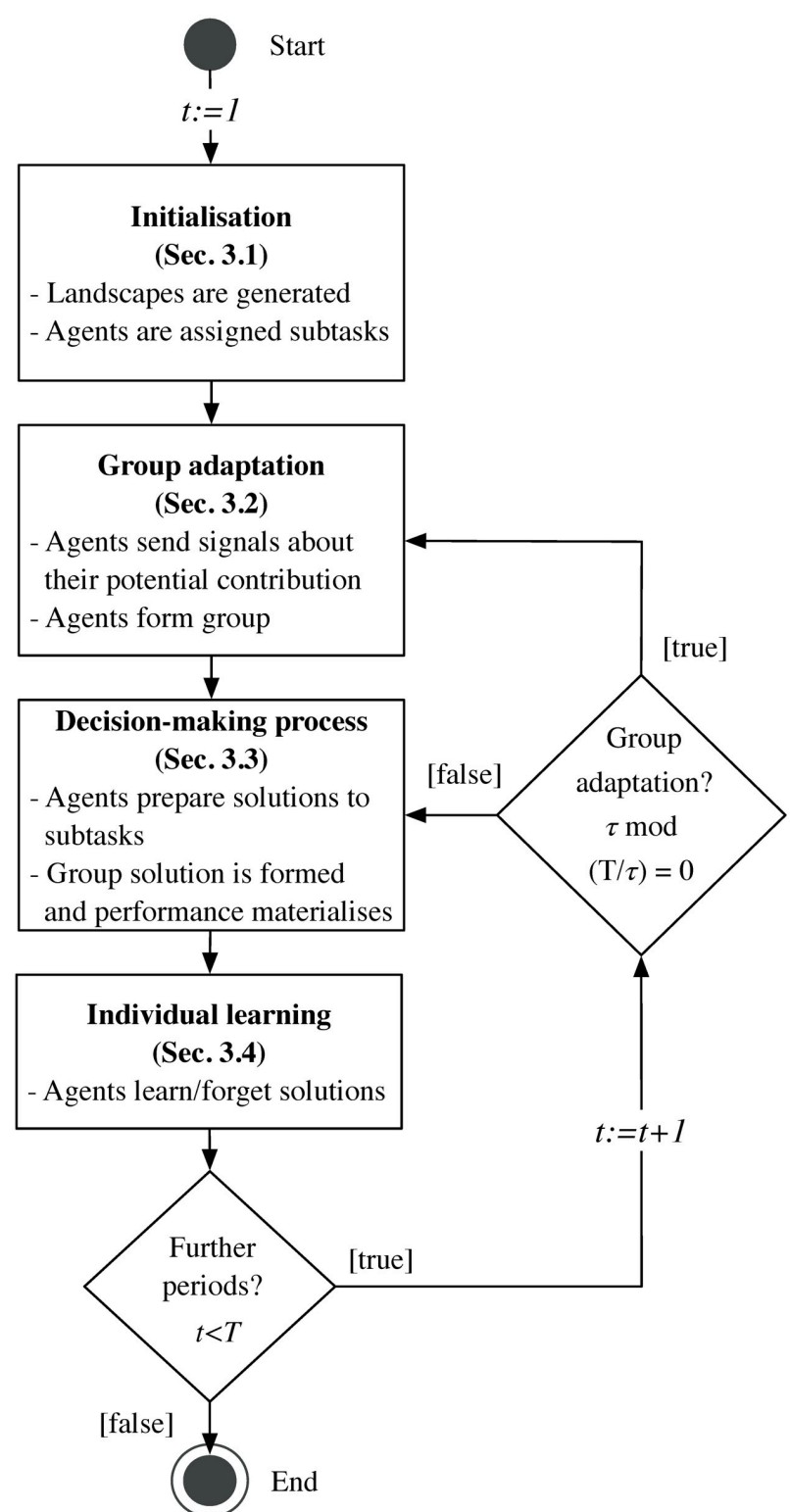

**Fig 1. Sequence of events during one simulation run.**

**Table 1. Parameters.**

| Type | Variables | Notation | Values |
|---|---|---|---|
| Independent variables | Task complexity | $K$ | {3, 5} |
| | Interdependence structure | *Matrix* | See Fig 2 |
| | Group adaptation | $\tau$ | {$\infty$, 1, 10} |
| | Learning probability | $\mathbb{P}$ | {0 : 0.1 : 1} |
| | Time steps | $t$ | {1 : 1 : 100} |
| Dependent variable | Task performance | $C(\mathbf{d_t})$ | [0, 1] |
| Other parameters | Number of decisions | $N$ | 12 |
| | Population of agents | $P$ | 30 |
| | Number of subtasks | $M$ | 3 |
| | Number of simulations | $\Phi$ | 1,500 |

## 3.1 Initialisation

**3.1.1 Task environment.** Following the *NK* framework [18], we model the complex task as a vector $\mathbf{d} = (d_1, \ldots, d_N)$ of $N$ binary decisions with $K$ interdependencies among them. Each decision $d_n \in \{0, 1\}$ contributes $c_n$ to group performance $C(\mathbf{d})$. The existence of interdependencies entails that the value of performance contribution $c_n$ depends on decision $d_n$ and $K$ other decisions, following

$$c_n = f(d_n, d_{i_1}, \cdots, d_{i_K}) , \tag{1}$$

where $\{i_1, \ldots, i_K\} \subseteq \{1, \ldots, n-1, n+1, \ldots, N\}$ and $0 \leq K \leq N-1$. The contributions are randomly drawn from a uniform distribution, $c_n \sim U(0, 1)$. The overall group performance $C(\mathbf{d})$ is the dependent variable in our analysis and is defined as the average of all performance contributions:

$$C(\mathbf{d}) = \frac{1}{N} \sum_{n=1}^{N} c_n . \tag{2}$$

Since the decisions are binary, there are $2^N$ possible solutions to the complex task. We compute the performance associated with the solutions according to Eq 2 and refer to the mapping of solutions to performances as the *performance landscape*. The parameter $K$ is an independent variable of the model (see Table 1), shaping the *complexity* of the task and, consequently, the ruggedness of the performance landscape. If decisions are not interdependent ($K = 0$), the performance landscape has a single peak. If decisions are interdependent ($K > 0$), by contrast, the landscape becomes more rugged, whereby high values of $K$ result in landscapes with various local maxima [18, 38]. These maxima correspond to solutions in which agents might get stuck, since there are no better alternatives in the neighbourhood (see Section 3.4) [18].

In this study, we model tasks of $N = 12$ dimensions, and divide them into $M = 3$ subtasks of equal length $S = N/M = 4$. We assume that one agent is sufficient to handle one particular subtask. Regarding task complexity, we consider tasks that are of either a low ($K = 3$) or a moderate complexity ($K = 5$). As tasks become more complex, i.e., as $K$ approaches $N - 1$, the performance landscape becomes more chaotic. This is why we do not consider tasks of high complexity (i.e., $K = N - 1 = 11$).

Furthermore, we distinguish six interdependence structures taken from [38]. The interdependence structure, which is one of the independent variables in our analysis (see Table 1),

reflects which performance contributions depend on which decisions [31, 39]. The interdependence structures included in this model can be described as follows.

- *Block*: Interdependencies are grouped into squares along the main diagonal. If tasks are of low complexity, they are perfectly decomposable into three independent subtasks. If tasks are moderately complex, reciprocal interdependencies between subtasks exist.

- *Centralised*: Interdependencies are located within the $K + 1$ first decisions. Consequently, when tasks are of low complexity, one agent highly affects the other agents' performance contributions. If tasks are moderately complex, the first agent influences the remaining agents' performance contributions, while the second agent influences part of the other agents' performance contributions. This interdependence structure characterises groups in which the power (to influence others) is unequally distributed.

- *Dependent*: In contrast to the centralised structure, the tasks assigned to one agent are highly dependent on the other agents' decisions. In a group setting, this interdependence structure indicates that there are two (more or less) independent subtasks, whose outcomes influence the third subtask.

- *Hierarchical*: The power to influence is concentrated at either one or two agents in the case of a low or a moderate complexity, respectively. Each decision influences the following contributions, but not the preceding ones.

- *Local*: Interdependencies are organised in a similar way as in a ring structure. If tasks are of low complexity, agents only affect their neighbours, i.e., for agents one and two, the interdependencies are located below the main diagonal. At the same time, there are interdependencies between the decision assigned to agent three and the performance of agent one.

- *Random*: Interdependencies are randomly located in the matrix.

Fig 2 illustrates the the six interdependence structures considered. The solid lines indicate the subtasks assigned to the agents.

**3.1.2 Agents.** We model a population of $P$ agents who are heterogeneous concerning their *capabilities*. They form a single group composed of $M < P$ members to solve a complex task. Agents are bounded in their rationality [40]: They want to maximise utility, but limitations in their capabilities constrain their behaviour. In particular, we limit their abilities in four ways:

1. We randomly assign each agent to one of the $M = 3$ areas of expertise with equal probability. Agents specialise in their assigned subtask, and they cannot solve any other subtask.

2. There are limitations in the agents' cognitive capacities. Consequently, they cannot oversee the entire solution space at a time. We denote the set of solutions known by agents at time $t$ by $\mathbf{S}_{mt}, = (\hat{\mathbf{d}}_{m1}, \ldots, \hat{\mathbf{d}}_{mI})$ where $\hat{\mathbf{d}}_{mi}$ represents a solution to subtask $\mathbf{d}_m$, $i = \{1, \ldots, I\}$ and $1 \leq I \leq 2^S$. Initially, agents are not aware of the entire set of $2^S$ solutions. Instead, at $t = 1$, each agent is endowed with one random solution $\hat{\mathbf{d}}_{mi}$ (i.e., $I = 1$). Agents learn about solutions to their subtask over time, and $\mathbf{S}_{mt}$ grows or shrinks as a result (see Section 3.4).

3. Agents aim to optimise their immediate utility. They do not consider the history of solutions implemented or their long-term utility when making their decisions.

4. Agents do not interact directly and cannot communicate with other agents, but there may be an indirect interaction. Specifically, the choices of one agent might affect the performance contributions of the remaining agents because of the interdependence structure (see Section 3.1.1).

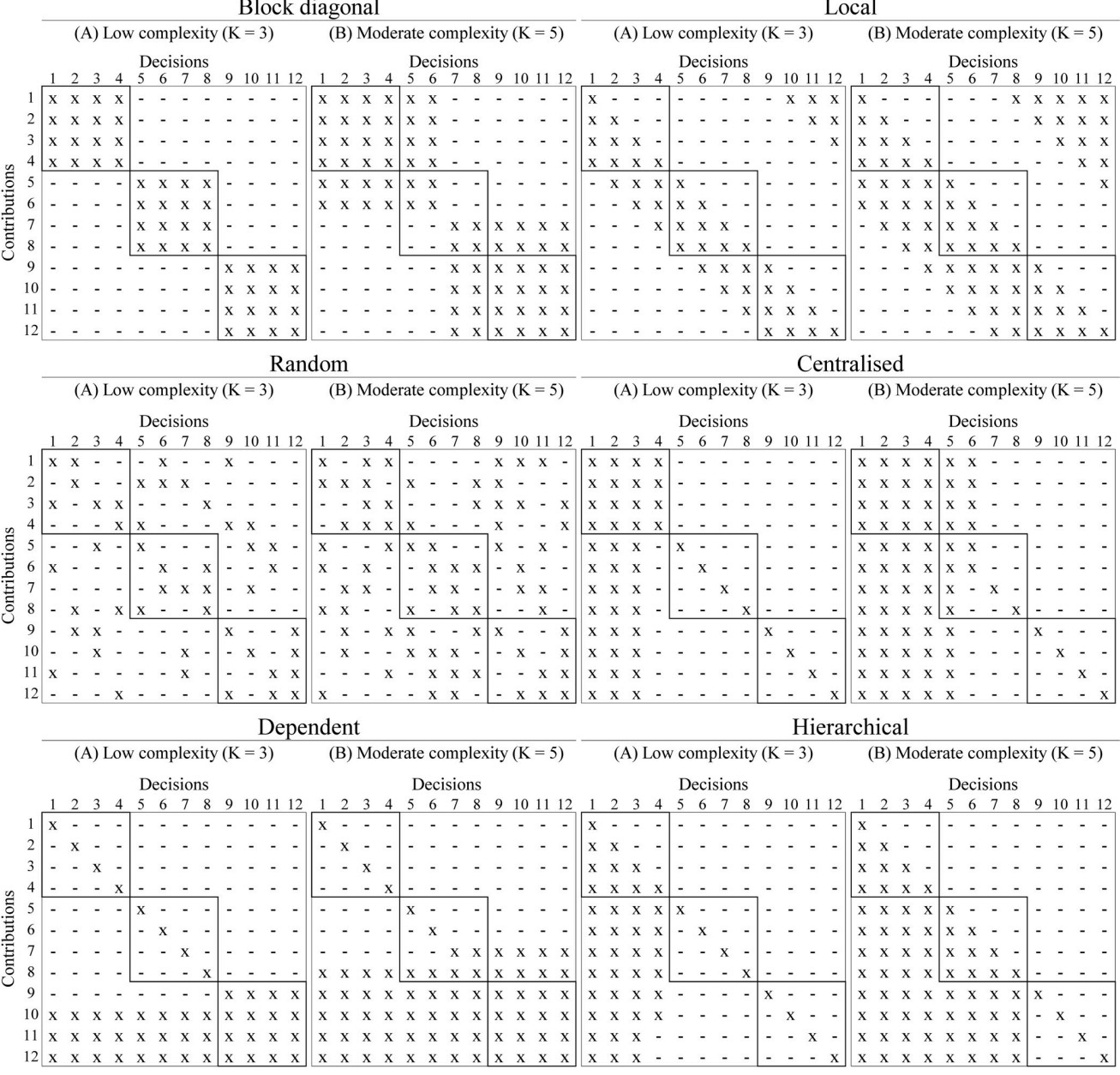

**Fig 2. Interdependencies between the performance contributions (represented on the y-axes) and the decisions (represented on the x-axes) are indicated with an *x*.** Each contribution depends on its own decision (see Eq 1), so there is an *x* in each element along the main diagonal. Solid lines indicate subtasks assigned to agents.

The utility of an agent assigned to the area of expertise *m* is the weighted sum of their performance contributions $C(\mathbf{d}_{mt})$ and the performance contributions of the remaining agents of the group $C(\mathbf{d}_{rt})$, where $r \in \{1, \ldots, M\}$ and $r \neq m$. The decisions located outside the scope of one agent are the *residual decisions*, and are denoted by $\mathbf{D}_{mt} = (\mathbf{d}_{1t}, \ldots, \mathbf{d}_{\{m-1\}t}, \mathbf{d}_{\{m+1\}t}, \ldots,$

$\mathbf{d}_{Mt}$). Agent $m$'s utility follows a linear function and is formalised in Eq 3:

$$U(\mathbf{d}_{mt}, \mathbf{D}_{mt}) = \frac{1}{2} \cdot \left( C(\mathbf{d}_{mt}) + \cdot \frac{1}{M-1} \sum_{\substack{r=1 \\ r \neq m}}^{M} C(\mathbf{d}_{rt}) \right) . \tag{3}$$

Agents receive positive utility only if they are part of the group. Non-member agents receive utility equal to 0.

## 3.2 Group adaptation

Once the parameters are set up, the agents form the group for the first time at $t = 1$. The group comprises $M = 3$ out of $P = 30$ agents, i.e., one agent per subtask. The objective of the group formation process is that the best-available experts join forces in a group. In this sense, a group that periodically adapts can integrate the agents with knowledge on the best-performing solutions within their ranks [10]. At $t$, agents estimate the utility for each solution they know, i.e., each solution in $\mathbf{S}_{mt}$. Recall that part of their utility comes from the residual decisions. Since we omit communication with other agents, they use the residual decisions implemented at the previous period $\mathbf{D}_{m\{t-1\}}$ as a basis for their estimations. The initial group solution is chosen randomly among the 4,096 different possibilities, as there is no decision-making process before the first period. Agent $m$'s *estimated utility* is then $U(\mathbf{d}_{mt}, \mathbf{D}_{m\{t-1\}})$.

After the agents have made their estimations, they compute the solution in $\mathbf{S}_{mt}$ that maximises their utility at time $t$ according to

$$\hat{\mathbf{d}}_{mt}^* := \arg\max_{\mathbf{d}' \subset \mathbf{S}_{mt}} U(\mathbf{d}', \mathbf{D}_{m\{t-1\}}) , \tag{4}$$

and send the signal $U(\hat{\mathbf{d}}_{mt}^*, \mathbf{D}_{m\{t-1\}})$. The signal can, for example, take the form of providing letters of application, disclosing their interest in a specific task, or completing questionnaires if applicable. The agents' signals are evaluated whenever a group is up to adapt. The agent who signals the highest estimated utility for a subtask $\mathbf{d}_m$ appears to be most suited to work on the subtask. Consequently, this agent joins the group and performs the assigned subtask. Non-member agents wait until the next group adaptation process occurs. If multiple agents send the same signal and this signals are the highest ones for a particular task, the agent who joins the group is chosen randomly from the top signalers.

Agents are assumed not to cheat and be honest and conscientious when giving signals. Also, agents always have an incentive to participate in the group since this is the only way to experience positive utility. Additionally, to avoid any strategic behavior when the agents make their estimations, the agents cannot observe the other agents' signals. Finally, we assume that all agents are fully aware of the group adaptation mechanism and its functioning.

The group adaptation process is repeated every $\tau$ periods. Thus, $\tau$ is an independent variable that reflects the frequency of group adaptation. The higher (lower) $\tau$ is, the less (more) frequently a group adapts. In this paper, we consider three different cases:

- *Long-term group composition*: The group is formed only once, i.e., at $t = 1$, and is stable throughout the observation period. For this case, we set $\tau = \infty$.

- *Medium-term group composition*: The group is formed at $t = 1$, and the group adaptation process takes place every $\tau = 10$ time steps.

- *Short-term group composition*: The group is formed at $t = 1$ and, subsequently, adapts in every time step. For this case, we set $\tau$ equal to 1.

### 3.3 Individual decision-making process and group strategy

Group members choose a particular solution to their assigned subtask at every time step $t$. Agents follow the decision-making rule described in Section 3.2. First, agents estimate the utility $U(\mathbf{d}_{mt}, \mathbf{D}_{m\{t-1\}})$ for each solution they know. Next, each agent computes and implements the solution with the highest estimated utility, i.e., $\hat{\mathbf{d}}^*_{mt}$ (see Eq 4), and the group strategy at time $t$ is formed by concatenating the solutions provided by all group members:

$$\mathbf{d}_t := \hat{\mathbf{d}}^*_{1t} \frown \cdots \frown \hat{\mathbf{d}}^*_{Mt} \; , \tag{5}$$

where $\frown$ is the concatenation of the solutions to all subtasks. We calculate task performance according to Eq 2, and the group members experience the resulting utility according to Eq 3. Finally, all agents—independent of whether they are group members or not—can observe the implemented group strategy, which serves as the basis for their decisions in the next period.

### 3.4 Individual learning

Recall that we initially endow each agent with one of the $2^S = 16$ solutions to their subtask. To overcome this limitation, agents have *learning capabilities* that allow them to explore the solution space. Learning occurs at the end of each time step $t$. Each of the $P = 30$ agents learns, even if they are not a group member. Learning consists of two independent mechanisms: *(i)* agents *discover* new solutions to their subtask, and *(ii)* agents *forget* solutions that are no longer relevant.

Regarding *(i)*, agents discover a new solution to their assigned subtask with probability $\mathbb{P}$. The new solution differs only in one bit from what the agents currently know, i.e., from the elements in $\mathbf{S}_{mt}$. Regarding *(ii)*, agents might forget a solution they know is not the utility-maximising solution at the current time step, i.e., not $\hat{\mathbf{d}}^*_{mt}$. If agents only know one solution (i.e., if $I = 1$), agents cannot forget this solution, because the only solution they know is the utility maximising solution at $t$ (i.e., $\hat{\mathbf{d}}^*_{mt}$). Forgetting occurs with the same probability $\mathbb{P}$ as learning does. We set equal probabilities to make it difficult for each agent to examine the complete solution space to their subtask [23, 24, 40, 41] and, thus, to appropriately consider the agents' limited capabilities (see Section 3.1.2). The probability $\mathbb{P}$ is an independent variable of our study and ranges between 0 and 1 in steps of 0.1.

Consequently, this learning mechanism allows agents to search for new, better-performing solutions to their subtask on the solution space, while discarding those solutions that may be useless. The sequential nature of this learning process implies that an agent's initial knowledge and past learning experience influences the solutions available in the future for this agent. Furthermore, this characterization of learning implies that the implementation of a group solution as described in Section 3.3 depends on the learning process of each individual agent.

## 4 Performance measures and analysis

To assure that the results are comparable across simulation runs and scenarios, we normalise the observed task performance by the maximum achievable performance. The normalised performance at each time step $t$ is computed according to $\bar{C}_t = C(\mathbf{d}_t)/C*$, where $\mathbf{d}_t$ represents the solution to the task implemented at time $t$, the function $C(\mathbf{d_t})$ computes the corresponding

performance (see Eq 2), and $C^*$ is the maximum achievable performance in that simulation round.

Furthermore, we perform a statistical analysis to investigate the relationship among the independent variables—the frequency of group adaptation, the probability of individual learning, task complexity, and the interdependence structure, see Table 1—and the dependent variable, task performance. Specifically, our aim is to unveil the functional relationship between the variables of interest and the normalised task performance. Our data analysis approach aligns with the argument brought forward in [42, 43]. They, amongst others, suggest employing regressions to analyse simulation data and parameter importance and to, finally, better understand the emergence of patterns. Using the Regression Learner feature in Matlab 2021b for our simulated data and following the minimum RMSE validation criteria [44], we train neural network regressions using the normalised performances. These RMSE and additional details on the regression models are reported in Table 2 of S1 Appendix.

Prior research has identified neural network models as useful alternatives to linear regressions when studying interrelations among multiple variables [45]. Neural network models are often categorized as *black-box models*, due to the difficulties in interpreting and assessing their predictions [46]. Conversely, neural network models allow researchers to avoid some of the most relevant restrictions of linear regressions, including the normality in errors assumption, sample sizes, the assumption of homocedasticity in the errors, and problems related to outliers and transformation of variables [45].

To properly interpret the results coming from neural network models, [47] suggests computing and reporting the partial dependencies of the dependent variable on a subset of the independent variables. One significant advantage of computing and visually representing the partial dependence functions of task performance on several independent variables is that it can offer a visual guide on the interrelations among the variables of interest [47].

Partial dependencies are calculated as follows. Let $\mathbf{X}$ be the set of all independent variables. The subset $\mathbf{X}^s$ includes either one or two independent variables that are in the scope of the analysis, and $\mathbf{X}^c$ consists of the remaining independent variables, i.e., the complementary set of $\mathbf{X}^s$ in $\mathbf{X}$. Then, $f(\mathbf{X}) = f(\mathbf{X}^s, \mathbf{X}^c)$ represents the trained regression model. The partial dependence of the performance on the independent variables in scope is defined by the expectation of the performance concerning the complementary independent variables so that

$$f^s(\mathbf{X}^s) = E_c(f(\mathbf{X}^s, \mathbf{X}^c)) \approx \frac{1}{V} \sum_{i=1}^{V} f(\mathbf{X}^s, \mathbf{X}^c_{(i)}) \ , \tag{6}$$

where $V$ is the number of independent variables in $\mathbf{X}^c$ and $\mathbf{X}^c_{(i)}$ is the $i^{th}$ element. By marginalising over the independent variables in $\mathbf{X}^c$, we get a function that depends only on the independent variables in $\mathbf{X}^s$. This partial dependence function allows to observe how task performance reacts on average to changes in the scope variable, i.e., the frequency of group adaptation, the individual learning probability, and task complexity in terms of the number and pattern of the interdependencies between subtasks.

## 5 Results and discussion

In this paper, we aim to understand better the effects of changing a group's composition more or less frequently ($\tau$) on performance. In addition, we aim at exploring the moderating effects of learning at the individual level ($\mathbb{P}$) and task complexity ($K$, *Interdependence structure*). Since we are interested in the average effects of these variables over time, we do not explicitly include time steps $t$ as a predictor in our analysis. In Section 5.1, we start by examining the overall effects of the frequency of group adaptation and the moderating factors. Next, we separately

examine the moderating factors. In particular, Section 5.2 provides insights into the moderating effect of learning at the agents' level, and Section 5.3 explores the moderating effect of the interdependence structure. Finally, Section 5.4 considers the two factors simultaneously and examines the resulting moderating effects.

## 5.1 Overall effects

In Fig 3, we present the partial dependencies between a predictor of interest and task performance at a time. Following Eq 6, this means that $E_c(f(\mathbf{X}^s, \mathbf{X}^c))$ is the estimated task performance and $\mathbf{X}^s$ is the independent variable chosen for analysis, i.e., the frequency of group adaptation $\tau$ in Fig 3A, the individual learning probability $\mathbb{P}$ in Fig 3B, the task complexity $K$ in Fig 3C, and the interdependence structure in Fig 3D.

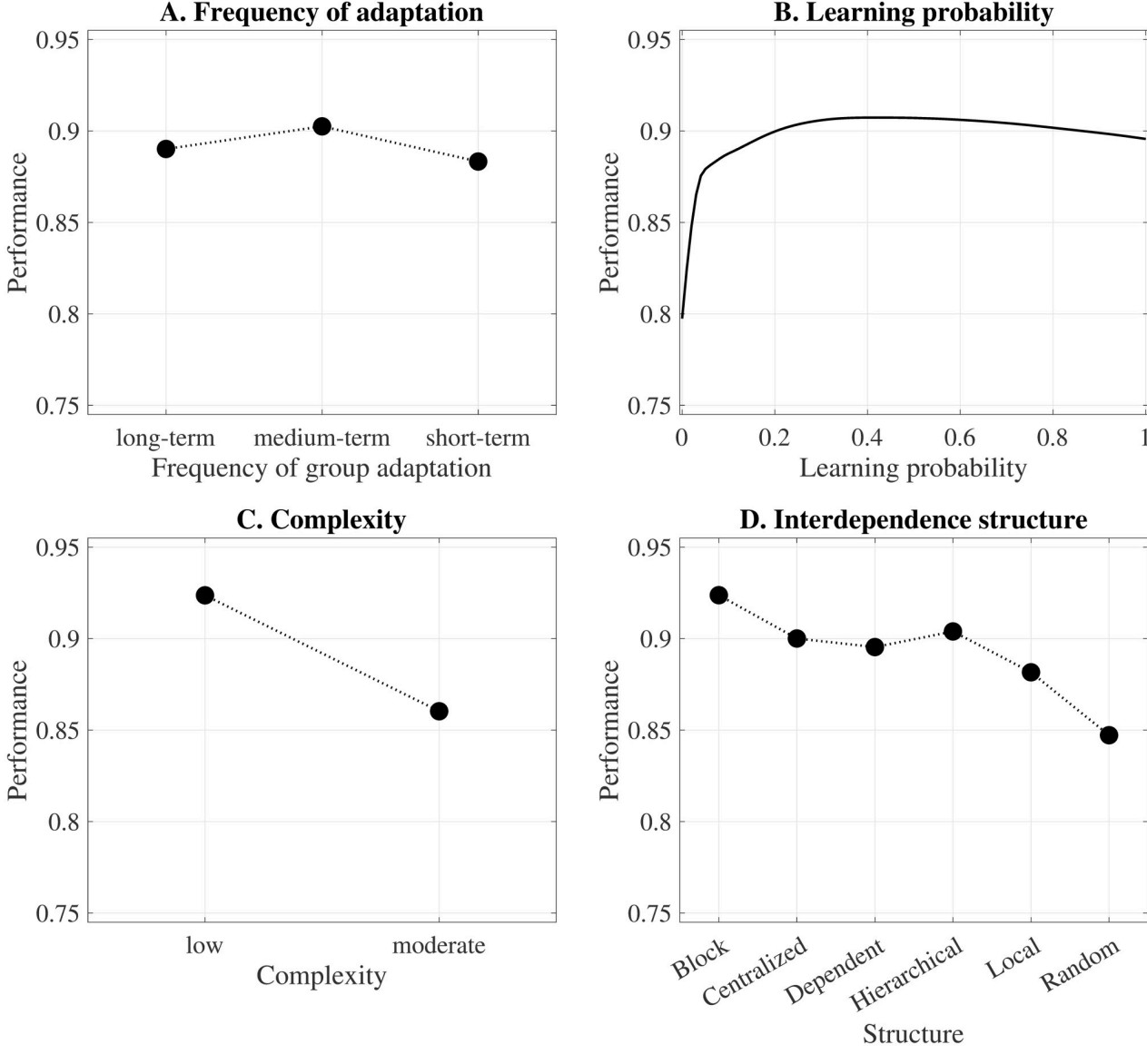

**Fig 3. Overall effects.**

Fig 3A shows that, compared to a long-term composition ($\tau = \infty$), the overall marginal effect of group adaptation on performance is only slightly stronger when groups are formed for the medium-term ($\tau = 10$) and slightly lower when groups are formed for the short-term ($\tau = 1$). The impact of group adaptation on task performance, thus, seems very small. We should note, however, that the partial dependence functions (see Section 4) only show the direct functional relationship between task performance and group adaptation, excluding the remaining factors from the representation. In the subsequent subsections, we investigate the role of the moderating factors, i.e., individual learning, task complexity, and the interdependence structure, to gain further insights into the nature of the relationship between task performance and group adaptation (see Sections 5.2 to 5.4).

Interestingly, as Fig 3B indicates, the functional relationship between the learning probability and task performance is more complex and relevant than the relationship between the frequency of adaptation and the performance. There is a strong positive effect when the learning probability is relatively low. For instance, between $\mathbb{P} = 0$ and $\mathbb{P} = 0.1$, performance grows on average from 0.7975 to 0.8865. However, this effect flattens when the probability is at intermediate levels. Between $\mathbb{P} = 0.1$ and $\mathbb{P} = 0.4$, performance only grows on average up to 0.9073. Finally, for higher learning probabilities, the slope turns slightly negative, even though one might expect that task performance always increases with individual learning. On average, at $\mathbb{P} = 0.7$, performance decreases to 0.9044 and to 0.8956 at $\mathbb{P} = 1$. Previous research argues that learning enables agents to master (challenging) task requirements, as it links consistent effort to discover efficient ways to solve a task [48]. The functional relationship between individual learning and task performance, however, contrasts with this interpretation. To further clarify the effect of learning in the relationship between group adaptation and task performance, we extend the individual learning analysis in Sections 5.2 and 5.4.

Regarding task complexity, Fig 3C shows that task performance decreases as complexity increases. Finally, Fig 3D suggests a relationship between the interdependence structure and task performance. In particular, the more pronounced cross-interdependencies between group members are, the lower task performance is. These findings are in line with previous research. [16, 18, 19], for example, argue that task complexity and the interdependence structure affect the performance landscape's ruggedness. Specifically, as subtasks become more interdependent—due to an increase in their number or to a dispersed structure—the performance landscape becomes more rugged. Since the global maximum is more difficult (easier) to find on rugged (single peaked) landscapes, complexity is negatively correlated with task performance. Sections 5.3 and 5.4 provide a more detailed analysis of this relationship.

## 5.2 Group adaptation and the moderating effect of the learning probability

The results in Section 5.1 suggest that individual learning has a more relevant impact on task performance than the frequency of group adaptation, especially when the probability of individual learning is low enough. Thus, this section analyses the overall effects presented in Section 5.1 in more depth by exploring how learning at the agents' level may impact the relationship between the frequency of group adaptation and task performance. To accomplish this, we run six neural network regressions. Regressions are performed for each of the frequencies of group adaptation and for each level of task complexity considered. Furthermore, we consider the probability of individual learning $\mathbb{P}$ between 0 and 1 in all regressions. Finally, to simplify the analysis, we only consider block interdependence structures. Table 2 in S1 Appendix provides a summary of these regressions. We calculate and plot the partial dependencies for the analysed scenarios in Fig 4A and 4B for tasks of low and moderate complexity, respectively. These figures show how, on average, group performance reacts to changes in the

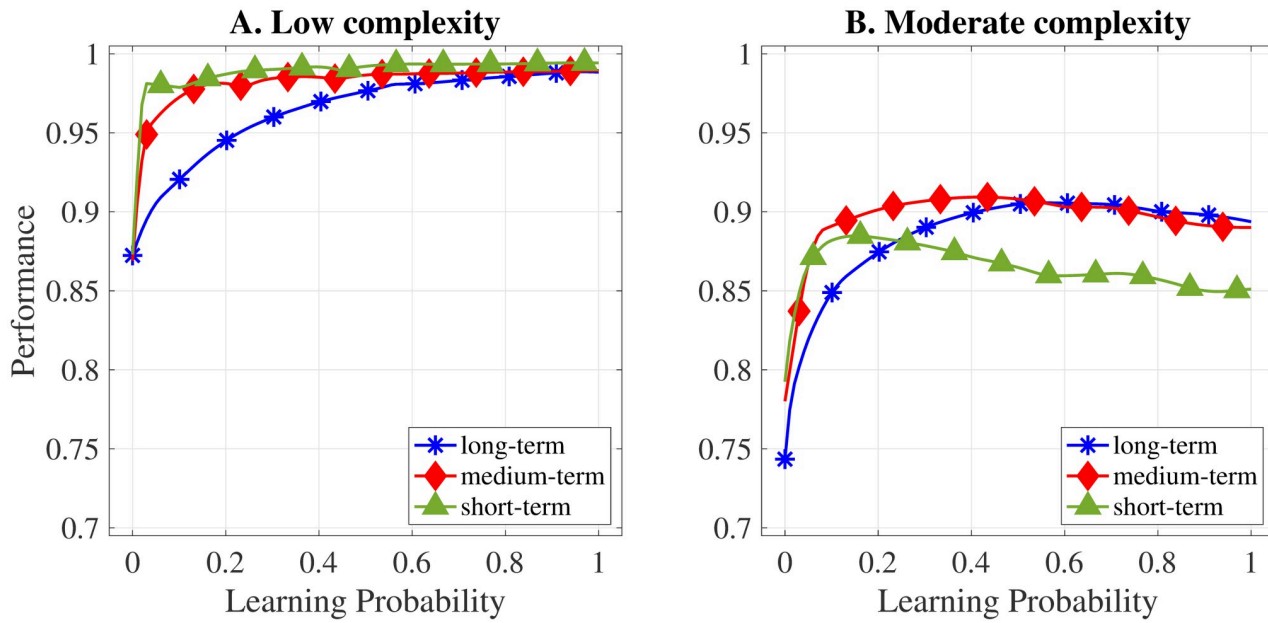

**Fig 4. Partial dependencies between task performance and the learning probability.**

probability of individual learning and how these changes differ depending on the frequency of group adaptation.

Results indicate that there is indeed a moderating effect of learning at the agents' level which changes the fundamental relationship between group adaptation and task performance. Specifically, the probability of individual learning affects task performance differently depending on the frequency of group adaptation. This is illustrated in Fig 4A and 4B, as the partial dependence functions of task performance on individual learning change depending on whether groups adapt periodically or not. For tasks of low complexity, groups of a medium- and short-term composition achieve similar performances for all learning probabilities and higher performances on average than groups of a long-term composition, i.e., groups which do not adapt. At relatively low levels of learning, task performance reacts more strongly to an increase in the learning probability when groups adapt frequently. Still, there are hardly any increases in performance beyond the threshold of 0.1. Conversely, the results indicate that the task performance of groups with a long-term composition increases steadily with each increase in the learning probability, although this positive effect decreases with each subsequent increase. Eventually, for learning probabilities above $\mathbb{P} = 0.7$, the performances converge to almost the same level for all three frequencies of group adaptation. Thus, our results suggest that long-term groups can compensate for the lower performance due to their lack of group adaptation by increasing the learning of their members.

For scenarios with moderately complex tasks, the results follow a different pattern (see Fig 4B). The partial dependence functions for all three scenarios of group adaptation exhibit somewhat similar shapes. Depending on the frequency of group adaptation, however, the impact of increasing the individual learning probability on task performance varies in magnitude and relevance. In all group adaptation scenarios considered, performance increases sharply when agents start to learn, i.e., when we move from $\mathbb{P} = 0$ to $\mathbb{P} = 0.1$. In contrast to low complexity scenarios, in which encouraging agents to learn never harms performance, an excessive learning probability might result in marginal adverse effects when tasks are moderately complex.

The tipping point, i.e., the learning probability at which the marginal effects of learning turn negative, depends on the frequency of group adaptation. In particular, the more frequently groups adapt, the lower is the learning probability at the tipping point. As a consequence of marginal adverse effects, groups of a short-term composition are worst off when the learning probability is higher than $\mathbb{P} \approx 0.25$. Conversely, this tipping point occurs at $\mathbb{P} \approx 0.4$ for medium-term groups and $\mathbb{P} \approx 0.6$ for long-term groups.

Regarding the effects of learning in groups and the interrelation between learning and group adaptation, previous theoretical research provides ambiguous interpretations on the issue. According to [49], creating competencies to solve tasks by promoting learning can be complemented by attracting new and competent group members. Prior experimental research supports these findings. For instance, [12, 13] show in their experimental study that group adaptation fosters creativity in problem-solving and increases task performance. By contrast, [50] argue that devoting excessive attention to exploratory processes—such as learning and group adaptation—might have negative consequences for task performance. Results of survey-based studies follow these insights and report a negative association between group adaptation and learning, as the former might offset the positive impact of the latter on task performance [7, 10, 11].

By employing an agent-based approach, we show that the results from prior research are complementary rather than conflicting. Specifically, we show in Fig 4A that groups which periodically adapt may gain from low and moderate levels of individual learning for tasks of low complexity. This aligns with the interpretation given by [49] of group adaptation and individual learning complementing each other. The positive impact of group adaptation on task performance, however, disappears if individual learning is high enough. In contrast, when tasks are moderately complex, frequent group adaptation may reduce the otherwise positive impact of individual learning, decreasing task performance (see Fig 4B). Our results, thus, are also consistent with the theoretical insights of [50]. Overall, results suggest that the the effect of group adaptation on task performance depends on the learning context. Groups that do not adapt frequently only rely on individual learning to obtain new knowledge. When learning is very low, these groups may struggle in finding new solutions to the task. In contrast, groups that adapt frequently may overcome these limitations by changing their composition, integrating new members within their ranks who possess knowledge not yet available in the group [51]. Conversely, when individual learning is high enough—and, thus, sufficient knowledge is available to the group—group adaptation is not advantageous in terms of performance.

## 5.3 Group adaptation and the moderating effect of the interdependence structure

This section extends the analysis of the overall effects presented in Section 5.1 by simultaneously considering variations in the interdependence structure (see Fig 2), task complexity, and the frequency of group adaptation. To simplify the analysis, we eliminate the effect of individual learning by fixing the probability to $\mathbb{P} = 0$. We run six neural network regressions (see Table 2 in S1 Appendix) and represent graphically in Fig 5A and 5B the partial dependencies of task complexity on the interdependence structure for each group adaptation and task complexity scenario.

When task complexity is low, the results show that the interdependence structure indeed has a moderating effect, especially in regards to how agents experience an influence from other agents. In the case of a block structure, there are no interdependencies across subtasks, the performance is the highest, and there are no differences in performance between frequencies of group adaptation. When there is a dependent structure, *one* agent is influenced from outside

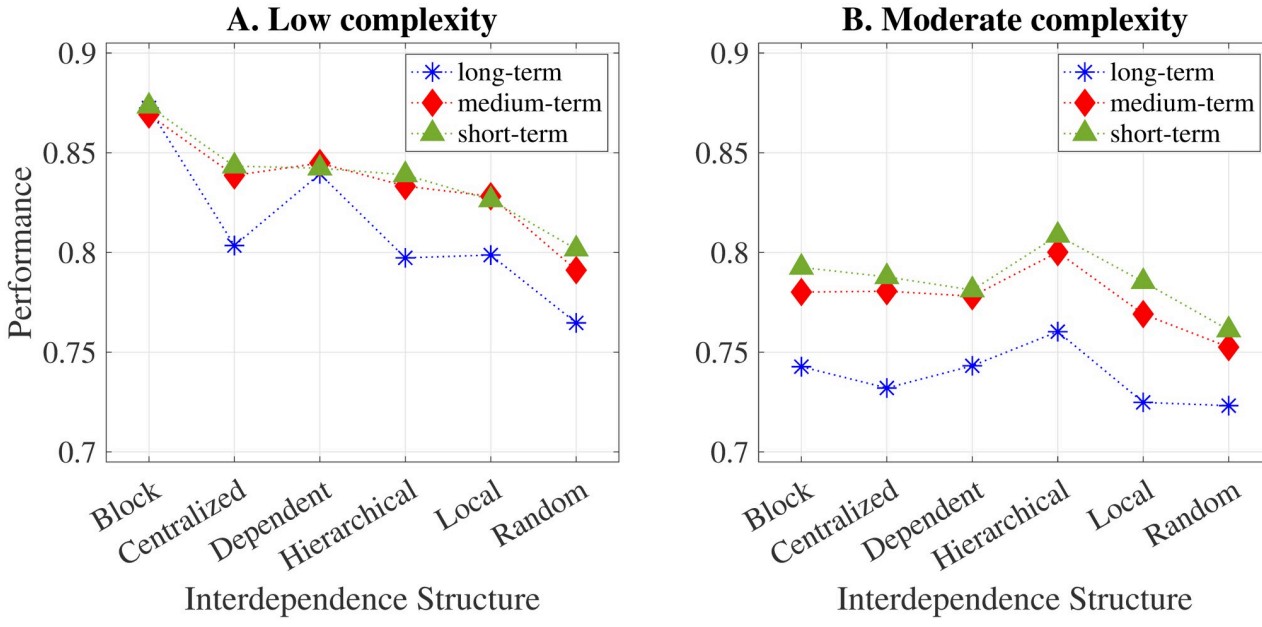

**Fig 5. Partial dependencies between task performance and interdependence structures.**

their area of responsibility. The performances are lower than in the case of a block structure, but there are no differences in performance between frequencies of group adaptation. In the cases of a centralised, hierarchical or local structure (see Fig 2), *two* agents are influenced by the other agents' decisions. In these cases, the following moderating effect unfolds: Groups that adapt ($\tau = 1$ or $10$) are better off than groups that do not adapt ($\tau = \infty$). It is also worth noting that it is almost negligible how often groups adapt since similar average performances can be observed for frequent and moderate adaptations. In the case of a random interdependence structure (see Fig 2), all *three* agents are influenced by the other agents' decisions. Again, we observe a relatively lower average performance for groups with a long-term composition. Conversely, groups with a short-term composition perform higher on average. Our results suggest that as interdependencies spread throughout the task, performance decreases, even if their number is relatively low. Thus, the number of agents who experience influence from other agents seems to be relevant for task performance. These results align with [19, 52], who argue that self-contained structures (the block pattern in our case) result in higher task performances, while interdependencies between subtasks come at the cost of performance.

In Fig 5B, we plot the partial dependencies for moderately complex tasks. In all scenarios considered, subtasks are interdependent (see Fig 2). In general, results for low complexity suggest that both the number and structure of interdependencies are equally relevant for determining task performance. Results for moderately complex tasks confirm this intuition. Since group members are interdependent in all cases considered, performance is substantially reduced. Groups which adapt frequently, however, are able to mitigate this negative impact comparatively more than long-term groups, and achieve higher performances in the process. Thus, our findings suggest that task complexity moderates the relationship between group adaptation and task performance. In particular, group adaptation leads to comparatively higher performances when interdependencies between agents are relatively pronounced—either due to an increase in the number of interdependencies or because they spread

throughout the task. This includes the centralised, hierarchical, local, and random structure in Fig 5A and all cases included in Fig 5B. Overall, this moderating effect increases the differences in task performance as compared to the general effects analysed in Fig 3A.

Controlling for task complexity and considering its interrelations with group adaptation over time with the help of an agent-based approach allows to avoid the usual limitations of prior research and to contextualise previous findings [5, 22, 24]. For example, [12, 13] argue that group adaptation positively affects performance because it fosters creativity. We show that these arguments only hold true for relatively complex situations (see Fig 5A and 5B). Additionally, our results suggest that groups adapt over time to the challenges posed by task complexity. Thus, our approach aligns with the arguments given by [5, 26], who claim that additional insights could be obtained by employing a longitudinal perspective on group adaptation. Recall that agents send signals about their estimations of what they can contribute to the group solution (see Section 3.2). If agents experience no or only minor influence from outside their area of responsibility, they can predict their contribution quite well. Consequently, it is assured that the most capable agents initially join forces in the group formed. Thus, when the agents' predictions are very precise, there are no reasons for further adaptation. However, if cross-interdependencies between agents exist, the accuracy of the agents' predictions decreases because of behavioural uncertainty. Consequently, there is a chance that *not* the most capable agents join forces when forming a group for the first time. Whenever the precision of the agents' predictions is low, repeated adaptations increase the chance that the most capable agents join forces in a group. This argumentation is in line with the findings presented in [41, 53, 54]. They argue that groups need to be heterogeneous to solve tasks efficiently. Also, the group members' skills should complement each other. Our agent-based approach allows to observe how groups *emerge* over time and that individuals who are best prepared for the task eventually join the group.

## 5.4 Group adaptation and simultaneous moderating effects

The analysis in this section considers a simultaneous variation in the learning probability and the interdependence structure and studies the resulting moderating effects. In Fig 6, we plot the partial dependencies for each frequency of group adaptation and for tasks of either a low or moderate complexity. Following Eq 6, this means that $E_c(f(\mathbf{X}^s, \mathbf{X}^c))$—the estimated task performance—depends on a vector of target variables $\mathbf{X}^s$ which contains *both* the interdependence structure and the probability of individual learning.

Fig 6A–6C presents the results for tasks of low complexity. In line with the results for a block interdependence structure presented in Section 5.2, increasing the learning probability up to 0.1 leads to an increase in performance. The more frequently groups adapt, the more strongly the performance reacts to increases in the learning probability at relatively low levels.

For the case of groups with a long- and medium-term composition and learning probabilities beyond 0.1, the marginal effects become less pronounced. Also, the performance reacts most strongly (weakly) to learning in the case of a block (random) structure, whereas the performances for the remaining structures converge to almost the same level. It is worth noting that the relative advantage of the dependent structure in the case of long-term groups and no learning (see Fig 6A) disappears with higher learning probabilities. Thus, most of the results for groups with a long- and medium-term composition and tasks of low complexity presented in Secs. 5.2 and 5.3 are robust against simultaneous variations in the learning probability and the interdependence structure.

On the contrary, high learning probabilities for groups with a short-term composition might result in marginal adverse effects. This is particularly the case for tasks with a centralised

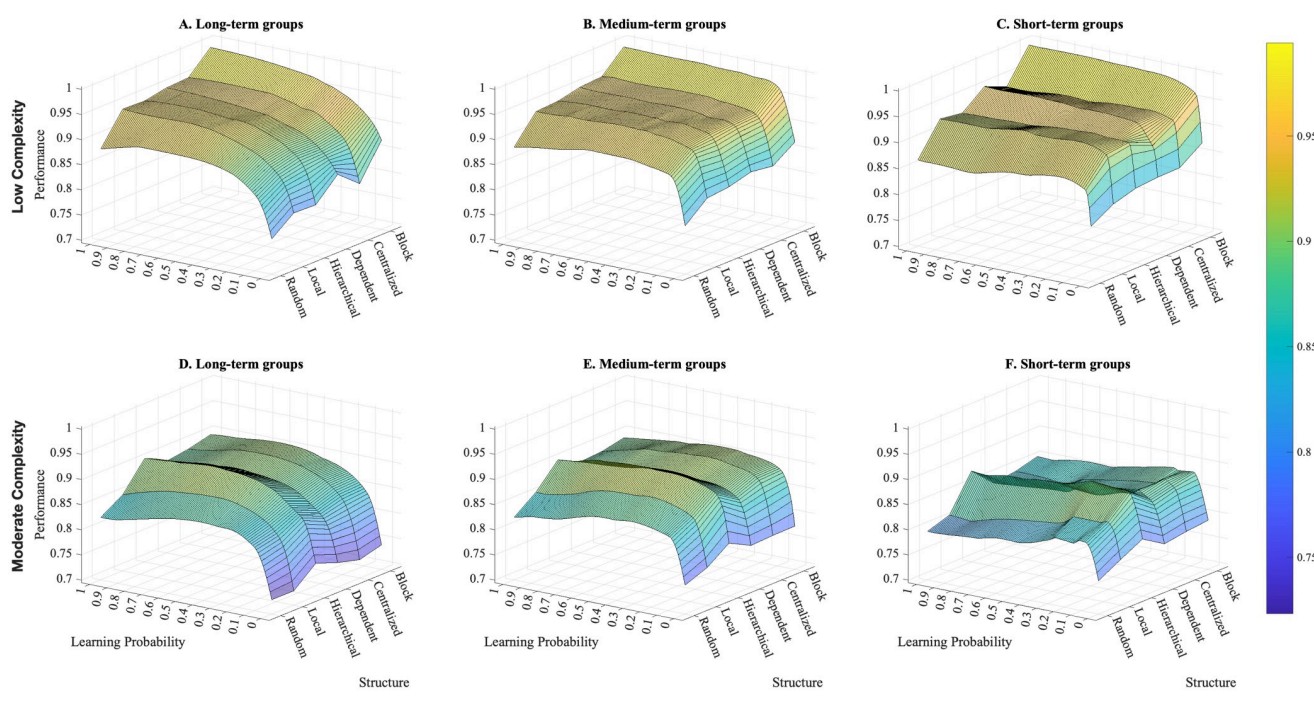

**Fig 6. Partial dependencies for a simultaneous variation of moderating factors.**

and hierarchical interdependence structure. Again, we observe the highest (lowest) performance for a block (random) interdependence structure. For the remaining interdependence structures, the effect of learning on performance stabilises beyond a learning probability of around 0.1.

The results for tasks of moderate complexity are presented in Fig 6D–6F. Recall, in Section 5.2 we focused on the block interdependence structure and found that learning substantially affects performance when groups adapt. The slopes of the surfaces plotted in Fig 6D–6F (in the range of a learning probability between 0 and 0.2) indicate that this finding is robust across all interdependence structures. Section 5.2 also found that increasing the learning probability beyond a certain threshold (tipping point) might unfold marginal adverse effects. Also, the tipping points are contingent on the frequency of group adaptation. The results for moderately complex tasks presented in Fig 6 confirm this finding for most interdependence structures. Only when interdependencies follow a hierarchical pattern, there appear to be no such (or less pronounced) marginal adverse effects. A possible reason is that interdependencies between agents are less pronounced for moderately complex tasks in this structure. For the remaining interdependence structures, the tipping points are at a learning probability of around 0.1 (see Fig 6F) and move to higher learning probabilities for groups of a medium- and a long-term composition (see Fig 6D and 6E).

Additionally, in Fig 6D–6F, we observe that the performances achieved by groups of a short-term composition are lower than those of the other groups for high learning probabilities. Recall, in Section 5.3 we found that long-term groups are worse off than groups of a medium- or short-term composition when there is no learning. This pattern shifts with an increase in the learning probability to moderate or high levels.

Results from prior experimental and survey-based research are often limited in the number of variables and do not consider simultaneous variations in the variables [24, 26]. Our agent-

based approach allows to analyse simultaneous variations and to specify the conflicting results found in prior research [10–13, 26, 27]. Our results also suggest that cross-interdependencies between agents appear to pose cognitive requirements on agents that come at the cost of learning efficiency and performance. This is in line with the (intrinsic) cognitive load theory [55]. Groups might avoid these issues by changing their composition, but this creates a risk of over-exploring at high levels of the individual learning probability [50].

## 6 Summary and conclusions

In this paper, we have implemented an agent-based model and ran simulations to analyse the effect of group adaptation on task performance. An agent-based modelling and simulation approach was chosen over other methods because it allows *(i)* to consider several variables at a time, i.e., group adaptation, individual learning and task complexity in terms of the number and pattern of interdependencies between subtasks, *(ii)* to control for simultaneous variations in these variables, and *(iii)* to take a longitudinal perspective on the effects of group adaptation on performance and on moderating effects. These advantages of the approach facilitate the generalisation of the results [24] and help specify the ambiguous results found in previous experimental, survey-based, and fieldwork research on group adaptation.

In particular, we show that changing a group's composition might indeed unfold positive effects on performance, but we also find that a more differentiated approach is required to manage groups efficiently. Overall, groups are always well-advised to adapt if they are low on learning. However, if groups are already high on learning, adaptation is not necessarily the most favorable strategy because adverse effects on performance are lurking. The task's interdependence structure and complexity reinforce both positive and negative impacts of learning on performance.

Although there are several advantages of an agent-based approach, there are also limitations. Compared to other research methods, i.e., in particular empirical methods, we model abstract agents instead of studying human decision-makers in a real-life setting. Additionally, we include assumptions in our model which are a simplification of reality. For example, we model human decision-makers who have perfect foresight when evaluating their performance landscape [56], who do not suffer from decision-making biases [57], and who can handle tasks of any complexity [58]—even though we only consider tasks of low and moderate complexity, but not of high complexity (i.e., $K = N - 1$). Moreover, we omit communication between agents and assume that the competencies developed by individual learning can easily be transferred between groups. These assumptions could be relaxed in future research. Future research might also include coordination mechanisms [30], skills that are (non-transferable) core competencies [59], and temporal developments [36]. Additionally, we focus on a relatively small group. Further research could analyze large-scale dynamics in organizations where agents vary in their experience. Finally, future studies could employ other research methods to contrast our findings, as suggested by [33].

## Supporting information

**S1 Appendix.**
(PDF)

**S1 File. Data from simulations.**
(XLSX)

**S2 File. Source code for simulations.**
(RAR)

**S3 File. Pre-print of prior manuscript version.**
(PDF)

## Author Contributions

**Conceptualization:** Darío Blanco-Fernández, Stephan Leitner.

**Data curation:** Darío Blanco-Fernández.

**Formal analysis:** Darío Blanco-Fernández, Stephan Leitner.

**Funding acquisition:** Stephan Leitner.

**Investigation:** Darío Blanco-Fernández, Stephan Leitner.

**Methodology:** Darío Blanco-Fernández, Stephan Leitner.

**Project administration:** Stephan Leitner.

**Resources:** Darío Blanco-Fernández.

**Supervision:** Stephan Leitner, Alexandra Rausch.

**Validation:** Darío Blanco-Fernández, Stephan Leitner.

**Writing – original draft:** Darío Blanco-Fernández, Stephan Leitner, Alexandra Rausch.

**Writing – review & editing:** Darío Blanco-Fernández, Stephan Leitner, Alexandra Rausch.

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
