## [Decision Letter · Decision Letter 0]

13 Jul 2023

PONE-D-23-09718The effects of group adaptation on task performance: An agent-based approachPLOS ONE

Dear Dr. Fernandez,

Thank you for submitting your manuscript to PLOS ONE. After careful consideration, we feel that it has merit but does not fully meet PLOS ONE’s publication criteria as it currently stands. Therefore, we invite you to submit a revised version of the manuscript that addresses the points raised during the review process.

We look forward to receiving your revised manuscript.

Kind regards,

Ayesha Maqbool, PhD

Academic Editor

PLOS ONE

Journal Requirements:

https://www.researchgate.net/publication/257094680_Learning_in_project-based_organizations_The_role_of_project_teams'_social_capital_for_overcoming_barriers_to_learning

In your revision ensure you cite all your sources (including your own works), and quote or rephrase any duplicated text outside the methods section. Further consideration is dependent on these concerns being addressed.

Reviewers' comments:

Reviewer's Responses to Questions

**Comments to the Author**

1. Is the manuscript technically sound, and do the data support the conclusions?

Reviewer #1: Yes

Reviewer #2: Partly

2. Has the statistical analysis been performed appropriately and rigorously? 

Reviewer #1: Yes

Reviewer #2: No

3. Have the authors made all data underlying the findings in their manuscript fully available?

Reviewer #1: Yes

Reviewer #2: Yes

4. Is the manuscript presented in an intelligible fashion and written in standard English?

Reviewer #1: Yes

Reviewer #2: Yes

5. Review Comments to the Author

Reviewer #1: The selection of research topic PONE-D-23-09718, "The effects of group adaptation on task performance: An agent-based approach” is very advanced and more applicable to existing scenarios. The author has attempted to describe the entire research intervention of a data-driven support model for group adoption and learning. During the process of reviewing, I found that there is merit in the research paper, and I have been accepted for publication by cross-checking and adding the following:-

1. Describe your model structure in a sequential, usual manner. State the random variable.

2. Group adoption of the task: This will depend on many attributes, such as whether the intelligent individual has adopted the task at an early stage, what measurement/ parameters you will consider for late adopters, and how to estimate its likelihood.

3.How does individual learning moderate the relationship between group adaptation and task performance? In this criteria, group adoption will depend on the interdependence among the many work policies. Please describe what measures you have taken to correlate the above stature.

4.How does complexity moderate the relationship between group adaptation and Task performance? The above criteria clearly depend on the above two indicators, as mentioned.

Reviewer #2: In the manuscript the statistical analysis of changing a group’s composition could have been performed appropriately and effects of changing a group’s composition could have been displayed with a relation.

6. PLOS authors have the option to publish the peer review history of their article (what does this mean?). If published, this will include your full peer review and any attached files.

Reviewer #1: **Yes: **Basavarajaiah DM

Reviewer #2: No

---

## [Author Response · Author response to Decision Letter 0]

9 Aug 2023

Preamble:

We are grateful for the detailed and thorough feedback on our paper, which helped us to further improve our work. Please see our detailed response of how we addressed the remarks and issues raised by the reviewers.

Summary of main changes:

• In response to the suggestions of Reviewer 1, we have added information in the model description (Section 3) to clarify the sequential order of the model and how it is related to the structure of the manuscript. Additionally, we have split subsection 3.5 into several parts, integrating its contents into sections 3.1-3.4. We hope this new model description structure aligns with the reviewer’s opinion.

• In response to the suggestions of the reviewers, we have separated the subsection which outlined the data analysis (i.e., Section) 3.6 from Section 3 and turned it into Section 4, adding additional explanations on the method. We hope this serves to clarify how the data analysis was performed, its justification, and the findings of the model.

• Following the reviewers’ suggestions, we have reviewed the results section (formerly Section 4, now Section 5) to clarify the relationship between the variables of interest (i.e., task performance, task complexity, the interdependence structure, individual learning, and group adaptation). We hope the relationship between these variables is now clearer.

• In the version of the manuscript with tracked changes (the Marked Version), the deleted text has been crossed out and changes are highlighted in green.

Best regards,

DARIO BLANCO-FERNANDEZ, STEPHAN LEITNER, ALEXANDRA RAUSCH

Detailed comments for Reviewer 1

Reviewer 1:

Describe your model structure in a sequential, usual manner. State the random variable.

Our answer:

We thank the reviewer for this comment. Following the reviewer’s suggestion, we have rearranged Section 3 (The Model) and added additional information to clarify the sequence of events of the model. This includes changing the introduction of the model (lines 141 to 169 in the Unmarked Version and 152 to 172 in the Marked Version).

Regarding the variables, we have rearranged the section and moved part of the content of Section 3.5 to the introduction to the model (Section 4, Table 1) and Section 3.1 (Figure 2). Additionally, we have clarified which are the main independent variables (group adaptation frequency, individual learning probability, task complexity, and interdependence structure) and the dependent variable (Task performance). 

Reviewer 1:

Group adoption of the task: This will depend on many attributes, such as whether the intelligent individual has adopted the task at an early stage, what measurement/ parameters you will consider for late adopters, and how to estimate its likelihood.

Our answer:

We thank the reviewer for this comment. Since agents engage in a sequential exploration process to learn about the complex task, their prior experience and initial positioning in the solution space influences the future availability of solutions for the agent (and hence the group). However, we do not address how agents learn to perform a task more efficiently over time. 

Reviewer 1:

How does individual learning moderate the relationship between group adaptation and task performance? In this criterion, group adoption will depend on the interdependence among the many work policies. Please describe what measures you have taken to correlate the above stature. 

Our answer:

Thank you very much for this comment. The effect of group adaptation is dependent on the learning context, as Fig. 4 shows. In particular, for lower levels of individual learning, groups that change their composition periodically perform comparatively better than stable groups. These differences are reduced as individual learning grows. Finally, for moderately complex tasks, performance might decrease if individual learning is sufficiently high. This decrease is larger, the more frequent group adaptation occurs.

To clarify this relationship, we have reworked Sections 5.1 and 5.2 (formerly Sections 4.1 and 4.2) to illustrate better which kind of moderating effect individual learning has. We have also added further explanations on how this relationship influences future group adoption of solutions in lines 699 to 706 in the Marked Version, and lines 509 to 516 in the Unmarked Version. Furthermore, we have extended the explanation of the statistical analysis in Section 4 (formerly Section 3.6) to make this relationship clearer. 

Reviewer 1:

How does complexity moderate the relationship between group adaptation and Task performance? The above criteria clearly depend on the above two indicators, as mentioned.

Our answer:

We thank you for your comment. Complexity moderates the relationship between group adaptation and task, as groups that adapt periodically mitigate the negative impact of task complexity (due to the number of structure of interdependencies) on task performance better than stable teams. This is illustrated in Fig. 5. As stated in the last paragraph of Sec. 5.3 (formerly 4.3), task complexity influences the adoption of solutions by groups because it prevents them from having the best-available composition from the beginning (i.e., when group members know the best-performing solutions at that moment). Group adaptation allows groups to reorganize their composition periodically, overcoming the problems caused by task complexity, and improving task performance in general. We have added additional explanations in Sec. 5.1 and 5.3 (formerly 4.1 and 4.3) to clarify this relationship.

Detailed comments for Reviewer 2

Reviewer 2: 

In the manuscript the statistical analysis of changing a group’s composition could have been performed appropriately and effects of changing a group’s composition could have been displayed with a relation.

Our answer:

Thank you very much for your comment. The statistical analysis performed in the paper follows Patel et al.’s (2018) and Law’s (2015) suggestions regarding simulation data analysis. To perform it, we relied on the Regression Learner feature in Matlab 2021b which, following the minimum RMSE validation criteria, suggested performing neural network regressions. RMSE and R2 were reported in Appendix B, Table 5.

Following prior literature on neural network regressions (Marquez et al, 1991; Wei Koh and Liang, 2017; Friedman, 2001), we represent the relationship between the variables of interest by calculating and plotting the partial dependence function on task performance on one or two target variables. Friedman (2001), in particular, recommends this method to present in a clear and simplified manner the results of neural network regressions.

To clarify the statistical analysis of this research, we have separated Section 3.6 to form Section 4 and extended the justification for the choice of the statistical analysis method. Furthermore, we have included explanations on neural network models and partial dependencies to add additional information on the statistical analysis of the paper. Furthermore, in the Results section (now Section 5, formerly Section 4) we have included additional explanations on the results depicted to clarify these aspects.

We hope your concerns are addressed with these additional explanations.

Detailed comments for the Editor

Editor: 

Our answer:

We thank you for your comment. We have reviewed the manuscript and adapted it to meet PLOS One’s requirements.

Editor: 

We noticed you have some minor occurrence of overlapping text with the following previous publication(s), which needs to be addressed. In your revision ensure you cite all your sources (including your own works), and quote or rephrase any duplicated text outside the methods section. Further consideration is dependent on these concerns being addressed.

Our answer:

Thank you very much for your comment. We have carefully reviewed the submitted manuscript and the reference provided, both manually and using anti-plagiarism software, and could not find the overlapping text. Moreover, the reference provided was already cited throughout the text. We have modified a small sentence in which the reference provided was already cited in case there was some error in the plagiarism check process. 

Editor: 

In your Data Availability statement, you have not specified where the minimal data set underlying the results described in your manuscript can be found. PLOS defines a study's minimal data set as the underlying data used to reach the conclusions drawn in the manuscript and any additional data required to replicate the reported study findings in their entirety. All PLOS journals require that the minimal data set be made fully available. For more information about our data policy, please see http://journals.plos.org/plosone/s/data-availability.

Upon re-submitting your revised manuscript, please upload your study’s minimal underlying data set as either Supporting Information files or to a stable, public repository and include the relevant URLs, DOIs, or accession numbers within your revised cover letter. For a list of acceptable repositories, please see http://journals.plos.org/plosone/s/data-availability#loc-recommended-repositories.

Our answer:

Thank you very much for this comment. We have updated the data availability statement to reflect that both the data and code are available at https://gitlab.aau.at/dablancofern/nk-model-for-dynamic-groups. We will add the data as a Supporting Information File to meet these requirements. Furthermore, we have uploaded the data, code, and figures to the following public repository: https://figshare.com/projects/The_effects_of_group_adaptation_on_task_performance_An_agent-based_approach/174876 and indicated so in the manuscript.

Editor: 

Please review your reference list to ensure that it is complete and correct. Any changes to the reference list should be mentioned in the rebuttal letter that accompanies your revised manuscript.

Our answer:

Thank you very much for your comment. We have reviewed the reference list. Some doi links have been deleted to keep consistency in the list. References have been reordered to consider the alphabetical order of authors. The following references have been added:

• Blanco-Fernández D, Leitner S, Rausch A. Interactions between the individual and the group level in organizations: The case of learning and group turnover. Central European Journal of Operations Research; 2023.

• Chai T, Draxler RR. Root mean square error (RMSE) or mean absolute error (MAE)? Arguments against avoiding RMSE in the literature. Geoscientific Model Development. 2014;7(3):1247–1250.

• Friedman J. Greedy function approximation: A gradient boosting machine. The Annals of Statistics. 2001;29(5):1189–1232.

• Marquez L, Hill T, Worthley R, Remus W. Neural network models as an alternative to regression. In: Proceedings of the Twenty-Fourth Annual Hawaii International Conference on System Sciences, IEEE; 1991. p. 129-135.

• Wei Koh P, Liang P. Understanding black-box predictions via influence functions. In: Proceedings of the 34th International Conference on Machine Learning; 2017;70. p. 1885-1894.

---

## [Editor Report · Decision Letter 1]

11 Aug 2023

The effects of group adaptation on task performance: An agent-based approach

PONE-D-23-09718R1

Dear Dr.Fernandez,

We’re pleased to inform you that your manuscript has been judged scientifically suitable for publication and will be formally accepted for publication once it meets all outstanding technical requirements.

Kind regards,

Ayesha Maqbool, PhD

Academic Editor

PLOS ONE
---

## [Editor Report · Acceptance letter]

17 Aug 2023

PONE-D-23-09718R1 

The effects of group adaptation on task performance: An agent-based approach 

Dear Dr. Blanco-Fernández:

I'm pleased to inform you that your manuscript has been deemed suitable for publication in PLOS ONE. Congratulations! Your manuscript is now with our production department. 

Kind regards, 

on behalf of

Dr. Ayesha Maqbool 

Academic Editor

PLOS ONE